# Meta-Research in Geriatric Surgery: Improving the Quality of Surgical Evidence for Older Persons in a Multidimensional-Scale Research Field

**DOI:** 10.3390/jcm13185441

**Published:** 2024-09-13

**Authors:** Ivan David Lozada-Martinez, David A. Hernandez-Paez, Isabela Palacios Velasco, Darly Martinez Guevara, Yamil Liscano

**Affiliations:** 1Biomedical Scientometrics and Evidence-Based Research Unit, Department of Health Sciences, Universidad de la Costa, Barranquilla 500366, Colombia; ilozada@cuc.edu.co; 2Center for Meta-Research and Scientometrics in Biomedical Sciences, Barranquilla 500366, Colombia; dhernandezp1@unicartagena.edu.co; 3Grupo Prometheus y Biomedicina Aplicada a las Ciencias Clinicas, School of Medicine, Universidad de Cartagena, Cartagena 130001, Colombia; 4Grupo de Investigación en Salud Integral (GISI), Departamento Facultad de Salud, Universidad Santiago de Cali, Cali 518300, Colombia; isabela.palacios00@usc.edu.co (I.P.V.); darly.martinez00@usc.edu.co (D.M.G.)

**Keywords:** aged, geriatric assessment, operative surgical procedures, evidence gaps

## Abstract

The world is facing a significant demographic transition, with a substantial increase in the proportion of older persons, as well as long-lived persons (especially nonagenarians and centenarians). One of the popular beliefs is that old age is synonymous with disease and disability. However, the successful aging hypothesis suggests that those older persons with advanced chronological age who maintain their functional capacity derive from it a delay in biological aging, enhancing the quality of organic aging and regulation. Therefore, regardless of chronological age, even in cases of extreme longevity, those older adults with a successful aging phenotype and favorable functional capacity would be expected to have satisfactory post-surgical recovery with a low risk of morbidity and mortality. Currently, there is a significant gap between the availability of high-certainty surgical evidence that allows for evidence-based interventions applicable to the long-lived population—taking into account the actual conditioning factors of the health phenotype in older persons—and, above all, predictors of satisfactory post-surgical evolution. The application of meta-research to geriatric surgery emerges as a fundamental tool to address this knowledge gap and reveals opportunities and limitations that need to be resolved in the near future to establish evidence-based surgical care for older persons. The aim of this manuscript was to present a real and globally relevant scenario related to surgical care, addressing the longevity, the availability, and the quality of surgical evidence applicable to this population, and also to present variables to consider in analysis and future perspectives in research and meta-research in geriatric surgery.

## 1. Introduction

Currently, the world is facing a significant demographic transition, with a substantial increase in the proportion of older persons, as well as long-lived persons (especially nonagenarians and centenarians) [1]. Considering that aging is a multidimensional, multilevel, and multidirectional process, the development of the health phenotype of older persons can vary over time and, in the worst case, take on a pathological aging phenotype linked to age-related chronic diseases and dependence [2]. There is often talk about the importance of controlling chronic diseases, as well as the rigorous implementation of pharmacological therapies to promote the survival of older persons [3]. However, there is usually resistance to the fact that older persons and long-lived persons continue to be exposed to the probability of developing an acute surgical disease or requiring a resolutive surgical procedure for a chronic disease [4], as well as the fact that it is a duty to provide them with the opportunity to access this service.

There is a significant gap between the availability of high-certainty surgical evidence that allows for evidence-based interventions applicable to the older adult population, taking into account the actual conditioning factors of the health phenotype in older persons, and, above all, predictors of satisfactory post-surgical evolution [5]. The application of meta-research to geriatric surgery emerges as a fundamental tool with which to address this knowledge gap and reveals opportunities and limitations that need to be resolved in the near future to establish evidence-based surgical care [6] for older persons and those with extreme longevity.

The aim of this manuscript was to present a real and globally relevant scenario related to surgical care, addressing the longevity, the availability, and the quality of surgical evidence applicable to this population, and also to present variables to consider for analysis and future perspectives in research and meta-research in geriatric surgery. 

## 2. Surgical Diseases in Older Persons and Extreme Longevity

With the increase in life expectancy globally over the preceding decades [7], the window of exposure to diseases at increasingly older ages has expanded, eventually elevating the risk of suffering a condition with potential need for surgical resolution. In older persons, the most common surgical diseases are directly related to the pathophysiological aspects of age-related chronic diseases, such as gallbladder stones or acute cholecystitis, new or recurrent hernias, and cancer [8,9].

Data from longitudinal cohorts in the United States have identified that the incidence of major surgery in older persons is 8.8 per 100 persons-year, with these mainly being elective surgeries. This incidence is higher among those aged 75 to 79 (10.8 per 100 persons-year) and in older persons with frailty (10.3 per 100 persons-year) [4]. Thus, the overall cumulative risk of major surgery over 5 years is 13.8%, with a rate of 12.1% for those aged 85 to 89, 9.1% for individuals aged 90 or older, and 12.1% for frail older adults [4]. Considering the epidemiological trends in population aging, it can be projected that the need for surgical resolution in older persons and those with extreme longevity may be significantly high, particularly in older adults with a pathological aging phenotype [10], characterized by a high prevalence of frailty and other debilitating conditions, such as sarcopenia, malnutrition, dementia, functional dependence, among others [10].

Evidently, with the advancement of science, technology, and innovation applicable to surgery, the use of novel techniques such as robotic surgery and the assistance of these techniques in minimally invasive surgeries will lead to major surgeries, which are currently not considered feasible for older persons [11], becoming routine in the near future. For this reason, the resolution of acute or chronic diseases through surgery in older adults will become increasingly frequent, and this requires the availability of the highest-quality, most cutting-edge evidence that takes into account the specific aspects of the older adult’s management in order to establish the best evidence-based surgical care.

## 3. Conception of Surgical Morbidity and Mortality Risk in the Longevity: Biological Age vs. Chronological Age

One of the popular beliefs is that old age is synonymous with disease and disability. Traditionally, it has been observed that decision-making in the healthcare of older persons is influenced by selective observation bias, which indirectly leads to a scenario of clinical ageism [12]. Making a priori judgments based on general health variables that are applicable in the surgical assessment of younger adults (such as the prevalence of chronic diseases) and attempting to extrapolate them to older persons is a serious mistake.

Approximately 15 years ago, Landi et al. [13] broke the paradigm of the belief that morbidity or multimorbidity were the best predictors of mortality in long-lived persons (aged 80 years or older). Through a longitudinal prospective cohort study with a four-year follow-up, these authors demonstrated that the effect of disability on the risk of death is greater, regardless of whether multimorbidity is involved (Hazard Ratio [HR] 3.91; 95% CI: 1.53–10.00) or not (HR 2.36; 95% CI: 0.63–8.83) [13]. From this point onward, various studies began to emerge that corroborated this trend [14,15], reshaping the traditional approach to older persons and recognizing the real value of specific outcomes that must be rigorously evaluated in determining the aging phenotype and health phenotype of older adults [16].

Starting from the fact that disability is a better predictor of morbidity and mortality in older persons [13], the assessment of functional independence, which directly correlates with biological age and not chronological age [17,18], takes on significant importance in the preoperative evaluation and prediction of post-surgical outcomes in older and long-lived adults [18]. The successful aging hypothesis suggests that those older persons with advanced chronological age who maintain their functional capacity (measured by their autonomy and independence in performing their general and specific daily activities) across various domains, such as physical, cognitive, emotional, and prosocial, among others, derive from it a delay in biological aging, reflecting the quality of organic aging and regulation [19]. Therefore, regardless of chronological age, even in cases of extreme longevity, those older adults with a successful aging phenotype and favorable functional capacity would be expected to have a satisfactory post-surgical recovery with a low risk of morbidity and mortality, independent of multimorbidity [18,19]. However, this approach should be interpreted with caution and, as in all medical disciplines, geriatric evaluation must be comprehensive and personalized, weighing benefits and risks.

The combined use of omics and translational studies to assess biological aging has allowed us to understand the mechanisms of organic adaptation and remodeling that explain the emergence of protective factors against accelerated aging, but also susceptibility to certain diseases [20]. It is believed that these same mechanisms explain the organic resilience that facilitates rehabilitation in this population. For this reason, it is important to highlight disruptive advances in the understanding of aging and the relevance of assessing multidimensional, multilevel, and multidirectional models of aging, which must necessarily include a preoperative geriatric and gerontological assessment [21,22]. This approach enables cohesion in multidisciplinary health decision-making by the clinical and surgical teams.

## 4. Clinical Evidence on Surgical Interventions in Longevity and Extreme Longevity: A Gap of Global Interest

According to clinicaltrials.gov, as of 21 August 2024, eight registered studies have included nonagenarians, of which two have directly or indirectly evaluated outcomes related to surgical interventions (assessing quality of life after open-heart surgery [NCT00248898], and revascularization in the management of critical limb ischemia [NCT02517840]). However, there is not one randomized controlled trial whose primary intervention and objective has been to evaluate the efficacy and safety of any surgical intervention in nonagenarians. The same scenario is observed in centenarians, where four studies are registered, but none are related to any surgical intervention or outcome, and no randomized controlled trial has been registered in this population either (Figure 1).

In octogenarians, specifically, 15 studies have been registered, of which 8 are directly or indirectly related to surgical outcomes. They are primarily grouped in relation to cardiovascular interventions (NCT02126202, NCT04252703, NCT02086019), oncological surgery (NCT03639792, NCT03904121, NCT06079229), and abdominal interventions (NCT04176432, NCT03813368). Only three randomized controlled trials have been registered, all of which evaluate the efficacy and safety of coronary revascularization (NCT02126202, NCT04252703, NCT02086019) (Figure 1).

As such, it is evident that there is a discrepancy between the increasing proportion of older persons and long-lived individuals globally, and the generation of new knowledge and the publication of clinical trials on surgical interventions that provide the highest-quality clinical evidence to address the potential health needs relating to surgical diseases that this population may present. This knowledge gap is notable, creating an important dilemma in evidence-based healthcare and surgical care for long-lived individuals. Therefore, it is necessary to promote proposals and research programs, focused on surgical interventions in older persons and extremely long-lived individuals, that address previous limitations and demonstrate the need and potential to promote disability-free life expectancy and overall life expectancy in long-lived persons who suffer acutely from surgical diseases.

When reviewing what has been published regarding surgical evidence in centenarians and nonagenarians (data as of 21 August 2024), it is noted that in 1955 the first publication was made, describing the outcomes of prostate surgery in centenarians. From that point on, up to 2024 (a window of 69 years), 637 and 54 scientific documents related to some type of surgical approach in nonagenarians and centenarians, respectively, were published in PubMed (Figure 2). Clearly, not all of these were original studies or involved samples exclusively made up of nonagenarians or centenarians. Many of the original studies consisted of cohorts of older persons that included a small proportion of nonagenarians or centenarians, limiting the ability to extrapolate their conclusions to this population. This would support the hypothesis of heterogeneity existing in surgical evidence in cases of extreme longevity, since, due to the scarcity of exclusive studies involving nonagenarians or centenarians, the specific criteria used during the comprehensive assessment and definition of the phenotype of extremely long-lived persons are biased, as geriatric and gerontological evaluation presents particular variations across different age groups, becoming more complex and rigorous in cases of extreme age [23].

## 5. Meta-Research in Geriatric Surgery: A Multidimensional Research Field to Strengthen Surgical Evidence

Previously, the need for checklists and guidelines that critically assess the quality of surgical evidence, which has distinct variables compared to non-surgical clinical evidence, has been discussed [24]. The same scenario occurs in the evaluation of methods in geriatric surgery. Considering the principles behind multidimensional, multilevel, and multidirectional models of aging, which become more intense at extreme ages [8,21], the design and execution of studies in geriatric surgery must adhere to standards and methodological foundations that can account for the particularities of this population. In other words, there should be a specific guide within the broader surgical evidence guidelines. 

Unlike other age groups, baseline outcomes must be identified to enable serious comparisons with minimal risk of bias. For example, evaluating dependent variables between healthy and unhealthy health phenotypes in long-lived persons could bias the interpretation of results and see either the overestimation or underestimation of conclusions in an attempt to extrapolate findings to the entire older population, which is not necessarily an accurate method [25]. Although there is no explicit definition with solid clinical–biological backing on the difference between healthy and unhealthy in extreme ages, it is undoubtedly not the same to have dementia or not, to have frailty or not, or to be functionally independent or not [26]. A strategy to obtain novel data that would allow for a deeper understanding of the differences in surgical research between long-lived persons and the group of older persons aged 60 to 80 years would involve designing translational surgical research studies that explore the pre- and post-surgical intervention responses, utilizing biological and molecular data. Therefore, when reporting these studies, a standardized checklist is needed to ensure the inclusion of essential characteristics for a deep understanding and accurate comparison of study groups involving older or long-lived persons.

In general, meta-research studies that evaluate the methodological quality, reporting, assessment, reproducibility, and incentives [27] ensuring the falsifiability of surgical evidence in older persons are scarce. Based on the emerging inclusion of new factors in multidimensional models of aging [16], it could be hypothesized that there is a significant gap in the prior research on certain geriatric and gerontological surgical outcomes [28]. However, this field has not been studied with great rigor.

Due to the lack of a precise understanding of the specific foundations of clinical and translational research [29], there is likely considerable heterogeneity in medical and biological definitions of older persons and extreme longevity (such as the age at which one is considered to be an older/long-lived person, why one is considered to be a healthy or unhealthy older/long-lived person, or the indiscriminate use of unvalidated geriatric or gerontological scales). The integration of non-clinical variables that are especially important in older persons, such as the quality of life, the self-perception of health, and subjective well-being [14,15,16], is relevant when making surgical intervention decisions for this population. It is essential to know, address, and recognize these considerations as hot topics in research. Taking into account these types of variables allows us to compare the certainty and long-term outcomes of the evidence generated, as opposed to studies that do not include them in their methods. Thus, meta-research in geriatric surgery is a currently unexplored multidisciplinary field with the potential to enhance the quality of surgical evidence and provide recommendations for the surgical care of older persons. 

## Figures and Tables

**Figure 1 jcm-13-05441-f001:**
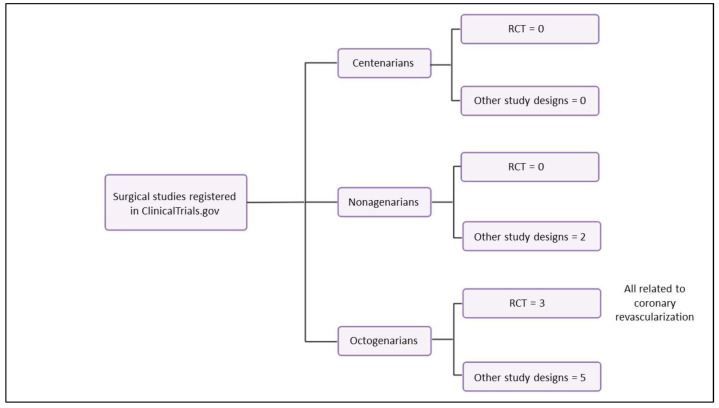
Surgical studies registered in clinicaltrials.gov related to long-lived persons.

**Figure 2 jcm-13-05441-f002:**
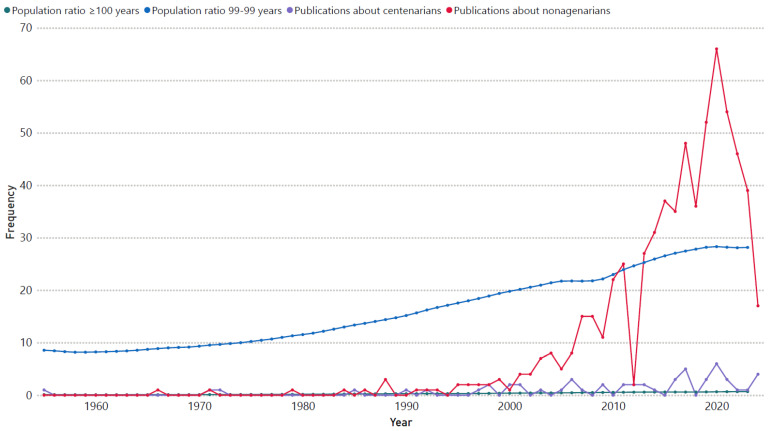
The relationship between the number of publications on surgical research in nonagenarians and centenarians in PubMed (data as of 21 August 2024), and the increase in the world’s long-lived population ratio over time. The long-lived population ratio was obtained by dividing the number of people in the specific age group (nonagenarians or centenarians) by the number of older persons (aged 65 or older) at the global level. Data were obtained from the UN’s World Population Prospects (2024). The chart created with power BI version 2.130.754.0.

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
