# Peer review of "Meta-Research in Geriatric Surgery: Improving the Quality of Surgical Evidence for Older Persons in a Multidimensional-Scale Research Field"

_jcm, 2024, doi:10.3390/jcm13185441_

Round 1
Reviewer 1 Report
Comments and Suggestions for Authors
Thank to the editors for the opportunity to review the work. This is a very interesting study on the problem of the lack of knowledge about the treatment of patients over 80 years of age. I agree with the authors that there is a large gap in the literature on this topic. This is related to the fact that there are no separate departments where such patients are treated, so far they are single people, and they disappear in publications concerning patients of different ages. However, this is a very important topic, due to the aging of society and the increasing need for EMB on the treatment of this group of patients.
The authors rightly noted that it is biological age, not chronological age, that influences the modification of treatment in patients. Therefore, it is important that the analyses of these patients are multidirectional. However, the work lacks a proposal on how we could obtain data on the treatment of such patients, as was proposed for the purpose of the study. I would ask the Authors to expand on this issue.
Although the nature of the study is prospective, I believe that in the review, which the authors undoubtedly performed, a detailed description of the methods they followed is necessary.
Author Response
We are grateful with the Editor and Reviewers for the expert reading of the manuscript. We have found the comments appropriate and helpful. They have permitted us to improve the manuscript. Appropriate changes were made and highlighted in the revised manuscript according to suggestions. Following are the responses to the comments.
Reviewer #1:
Thank to the editors for the opportunity to review the work. This is a very interesting study on the problem of the lack of knowledge about the treatment of patients over 80 years of age. I agree with the authors that there is a large gap in the literature on this topic. This is related to the fact that there are no separate departments where such patients are treated, so far they are single people, and they disappear in publications concerning patients of different ages. However, this is a very important topic, due to the aging of society and the increasing need for EMB on the treatment of this group of patients.
---The authors rightly noted that it is biological age, not chronological age, that influences the modification of treatment in patients. Therefore, it is important that the analyses of these patients are multidirectional. However, the work lacks a proposal on how we could obtain data on the treatment of such patients, as was proposed for the purpose of the study. I would ask the Authors to expand on this issue.
R/ We thank the reviewer for this comment. In the second paragraph under the subtitle of “Meta-research in geriatric surgery: a multidimensional research field to strengthen surgical evidence”, the following is presented: “Unlike other age groups, baseline outcomes must be identified to enable serious comparisons with the least risk of bias. For example, evaluating dependent variables between healthy and unhealthy health phenotypes in long-lived persons could bias the interpretation of results and either overestimate or underestimate conclusions, at-tempting to extrapolate findings to the entire older population, which is not necessarily accurate [25]. Although there is no explicit definition with solid clinical-biological backing on the difference between healthy and unhealthy in extreme ages, it is undoubtedly not the same to be demented or not, to have frailty or not, or to be function-ally independent or not [26]. Therefore, when reporting these studies, a standardized checklist is needed to ensure the inclusion of essential characteristics for a deep understanding and accurate comparison of study groups among older or long-lived persons”. The above allows for the appreciation of methodological aspects that must be thoroughly evaluated prior to designing a surgical research study on extreme longevity, in order to obtain data with adequate internal and external validity. However, we supplement this paragraph with the following statement, addressing the reviewers' concerns:
A strategy to obtain novel data that would allow for a deeper understanding of the differences in surgical research between long-lived persons and the group of older persons aged 60 to 80 years would involve designing translational surgical research studies that explore the pre- and post-surgical intervention responses, utilizing biological and molecular data.
---Although the nature of the study is prospective, I believe that in the review, which the authors undoubtedly performed, a detailed description of the methods they followed is necessary.
R/ We thank the reviewer for this comment. Considering that this manuscript is a perspective, we are unclear about what is meant by a deepening of the methods, given that this analysis does not address a specific research question that would require a detailed methodology. The only original analyses were those conducted and presented in the text and in the figure legends.

Reviewer 2 Report
Comments and Suggestions for Authors
Dear authors, the topic you have chosen is very interesting, the structure is well organized and understandable, the language used appropriately.
Just a few questions: Which area of ​​geriatric surgery should we focus on? Where are there any substantial differences with middle-aged people?
Author Response
We are grateful with the Editor and Reviewers for the expert reading of the manuscript. We have found the comments appropriate and helpful. They have permitted us to improve the manuscript. Appropriate changes were made and highlighted in the revised manuscript according to suggestions. Following are the responses to the comments.
Dear authors, the topic you have chosen is very interesting, the structure is well organized and understandable, the language used appropriately.
Just a few questions: Which area of geriatric surgery should we focus on? Where are there any substantial differences with middle-aged people?
R/ We thank the reviewer for this query. Given the increasing healthcare demands of the long-lived persons, which will continue to grow over time, and considering the incidence of surgical diseases highlighted in the section 'Surgical Diseases in Older Persons and Extreme Longevity,' it can be argued that abdominal surgery and oncologic surgery may be the two areas of geriatric surgery most in need of support.
Unlike middle-aged individuals, long lived-persons—depending on their health phenotype based on geriatric and gerontological outcomes—reflect the course of biological aging and the organic responses (1). The current hypothesis is that long-lived persons, by the fact of their extreme longevity, exhibit unique characteristics that provide protection against exposome. In cases where successful aging has occurred, their health phenotype in old age, regardless of chronological age (2), suggests they may still exhibit favorable biological responses to clinical or surgical interventions, without age being a determining factor (2).
- Liu WS, You J, Ge YJ, Wu BS, Zhang Y, Chen SD, et al. Association of biological age with health outcomes and its modifiable factors. Aging Cell. 2023; 22(12):e13995.
- Quero G, Pecorelli N, Paiella S, Fiorillo C, Petrone MC, Capretti G, et al. Pancreaticoduodenectomy in octogenarians: The importance of "biological age" on clinical outcomes. Surg Oncol. 2022; 40:101688.
We thank the reviewer for careful reading of the manuscript and pertinent correction and suggestions. This question is answered with the argument of the previous question.

Round 2
Reviewer 1 Report
Comments and Suggestions for Authors
Thank the Authors for the answers. They responded to the questions explaining the doubts.